# Study on Few-Shot Fault Diagnosis Method for Marine Fuel Systems Based on DT-SViT-KNN

**DOI:** 10.3390/s25010006

**Published:** 2024-12-24

**Authors:** Shankai Li, Liang Qi, Jiayu Shi, Han Xiao, Bin Da, Runkang Tang, Danfeng Zuo

**Affiliations:** 1School of Automation, Jiangsu University of Science and Technology, Zhenjiang 212100, China; 221210301111@stu.just.edu.cn (S.L.);; 2Jiangsu Shipbuilding and Ocean Engineering Design and Research Institute, Zhenjiang 212100, China

**Keywords:** marine fuel system, few-shot fault diagnosis, Siamese network, transformer, KNN

## Abstract

The fuel system serves as the core component of marine diesel engines, and timely and effective fault diagnosis is the prerequisite for the safe navigation of ships. To address the challenge of current data-driven fault-diagnosis-based methods, which have difficulty in feature extraction and low accuracy under small samples, this paper proposes a fault diagnosis method based on digital twin (DT), Siamese Vision Transformer (SViT), and K-Nearest Neighbor (KNN). Firstly, a diesel engine DT model is constructed by integrating the mathematical, mechanism, and three-dimensional physical models of the Medium-speed diesel engines of 6L21/31 Marine, completing the mapping from physical entity to virtual entity. Fault simulation calculations are performed using the DT model to obtain different types of fault data. Then, a feature extraction network combining Siamese networks with Vision Transformer (ViT) is proposed for the simulated samples. An improved KNN classifier based on the attention mechanism is added to the network to enhance the classification efficiency of the model. Meanwhile, a Weighted-Similarity loss function is designed using similarity labels and penalty coefficients, enhancing the model’s ability to discriminate between similar sample pairs. Finally, the proposed method is validated using a simulation dataset. Experimental results indicate that the proposed method achieves average accuracies of 97.22%, 98.21%, and 99.13% for training sets with 10, 20, and 30 samples per class, respectively, which can accurately classify the fault of marine fuel systems under small samples and has promising potential for applications.

## 1. Introduction

Currently, the majority of vessels globally are propelled by diesel engines [1]. In contrast to other shipboard equipment, diesel engines consist of various subsystems, including the fuel system, lubricating oil system, and cooling water system [2]. These subsystems exhibit varying probabilities of fault due to their distinct functions and operating conditions. An analysis of shutdown faults provided by the Association of British Diesel Engine Engineers and Users indicates that fuel system faults account for 60% of the total diesel engine faults, which represents the highest proportion [3]. Consequently, research on fault diagnosis for the fuel system is pivotal for enhancing the reliability of diesel engines and ensuring the safe navigation of vessels.

Digital twin, as an emerging virtualization method, enables more accurate fault diagnosis by building virtual models of physical entities for simulated computation and deduction [4]. The scholarly community has initiated a series of research on the application of DT for fault diagnosis. Liu et al. [5] leveraged structural digital twin technology to generate simulated monitoring signals and as an input to the designed network, significantly enhancing its performance of fatigue damage detection across different structures. Xu et al. [6] developed a DT model of a centrifugal pump, which was used to simulate a spectrum of health conditions and augmented the dataset, finally constructing a graphical convolutional neural network, and facilitating cross-domain fault diagnosis. Yi et al. [7] made a significant stride by integrating a bearing dynamics model within a neural network architecture. Their innovative method involved using actual vibration data to construct a DT model of the bearing, thereby enriching the fault repository and enhancing the diagnostic accuracy. Liu et al. [8] analyzed faults under constant and variable speed conditions by establishing a fault diagnosis test platform. Then, the obtained data are integrated into the Unity3D platform to realize online diagnosis and the digital twin model is conducted for subsequent fault diagnosis of reducer. Despite the notable achievements of DT technology in these areas, its application in diesel engine fault diagnosis still needs to be explored.

Alternatively, deep learning has achieved excellent results in the field of fault diagnosis due to its ability to extract data features and use them fully [9]. Nonetheless, deep learning-based diagnostic methodologies typically necessitate an extensive array of fault samples, which is difficult to realize in practice due to many constraints. Few-shot learning provides a solution to this problem [10,11], enabling models to learn new tasks or categories with few labeled samples quickly. At present, most few-shot learning methods are based on data augmentation [12], transfer learning [13], and meta-learning [14], with researchers worldwide actively engaged in exploring these approaches. Li et al. [15] introduced a generalized few-shot classification framework founded on Convolutional Neural Networks (CNN). This framework innovatively incorporates an orthogonal Softmax layer as the network’s classifier, thereby maximizing the distinction between training and test samples, optimizing the network’s architecture, and achieving structural simplification. Li et al. [16] proposed a fault diagnosis method which converts raw vibration signals into two-dimensional time-frequency images and extracts deep features using KANs-CNN. Then, the FAN module aggregates features from multiple levels, and data generation through diffusion networks addresses the small sample issue. Fu et al. [17] proposed a few-shot fault diagnosis method based on Deep Auto-Encoder (DAE) and transfer learning. This approach pre-trains a DAE on a normal sample corpus, then applies SVM classification to fault samples via transfer learning, as proven effective on a civil aviation dataset. Despite these advancements, which have enhanced diagnostic accuracy in scenarios with small samples, there remain certain challenges. For instance, data augmentation may introduce noise and other negative effects during the generation process. Transfer learning not only relies on a large amount of source domain data but also requires that the data distribution difference between the source and target domains should not be too large. Meanwhile, meta-learning requires too many computational resources.

Compared with the above methods, the Siamese network, as a comparison network, known for its simple architecture and ease of training, is extensively applied in fields such as image processing [18], fingerprint recognition [19], target tracking [20]. During training, two samples are randomly selected and input into the network for training, when the number of training samples is *n*, a total of $C_{n}^{2}$ times of effective training for the network, so in the case of small samples, the Siamese network can expand the number of training times of the network, which effectively solves the problem of network underfitting due to the lack of training samples [12]. However, the utilization of Siamese networks also entails a suite of considerations that require resolution:

(1) Common Siamese networks mostly use CNNs as their subnetworks, which are limited by the influence of the receptive field; CNNs can only extract local features and cannot focus on the global information of the input samples.

(2) The Siamese network is a comparison network; its output is the degree of similarity of the two input samples. Consequently, when confronted with multiclass classification tasks, the samples to be tested need to be compared with all types of samples one by one; this approach inherently diminishes classification efficiency.

In response to the challenges previously mentioned, this paper introduces a novel DT-SViT-KNN few-shot fault diagnosis method. Initially, a diesel engine DT model is constructed by integrating the mathematical model, mechanism model, and three-dimensional physical model of the Medium-speed diesel engines of 6L21/31 Marine. Then, the DT model is used to calculate fault simulation to obtain different types of fault samples. For the acquired fault samples, this paper proposes a feature extraction network that merges Siamese network architecture with ViT. Finally, an improved KNN classifier based on the attention mechanism is added on the basis of this network to improve the model classification efficiency. The specific contributions of this paper are delineated as follows:

(1) This study constructs a comprehensive DT model that integrates diesel engine mathematical, mechanical, and three-dimensional physical models, which facilitates the mapping from the physical entity to its virtual counterpart. In conjunction, a novel feature extraction network is devised by combining the Siamese network with the ViT, enabling the extraction of global multiscale features from the input data.

(2) For the issue of the model’s limited discriminative capacity between similar sample categories, we designed a Weighted-Similarity loss function. This innovation introduces similarity labels and penalty coefficients, thereby enhancing the model’s efficiency in distinguishing between similar and dissimilar samples. This advancement elevates the differentiation accuracy of similar and distinct sample pairs to a novel level of precision.

(3) The incorporation of an attention mechanism during the KNN search phase allows the model to account for the relevance and significance of each neighbor rather than relying solely on distance-based classification. The attention-scoring mechanism enables the model to assign greater weight to more pertinent samples, enhancing overall classification accuracy.

The rest of this paper is organized as follows: Section 2 presents the detailed theoretical information of the proposed method. Experimental validation is carried out in Section 3. Section 4 gives the conclusion.

## 2. Proposed Method

Figure 1 shows the overview of the fault diagnosis method proposed in this paper, which consists of diesel engine entity, mathematical model, twin model, and state identification. On the basis of the diesel engine entity, the diesel engine DT model is established. Through the diesel engine DT model for fault simulation to obtain different types of fault data, the system uses the fault diagnosis algorithm for fault diagnosis and outputs the diagnosis results.

### 2.1. Diesel Engine Digital Twin Model Creation

#### 2.1.1. Mathematical Model Construction of Diesel Engine

The mathematical model of a diesel engine can be divided into the following parts by components [21]:

(1) Diesel engine body

The diesel engine dynamics equations are as follows:(1)$$\frac{\pi I}{30}\cdot \frac{dN_{D}}{dt}=M_{d}-M_{l}$$
(2)$$M_{d}=C_{0}\cdot g_{cyl}\cdot \eta_{cy}-M_{f}$$
where $M_{f}$ represents the friction loss of the diesel engine, $\eta_{cy}$ represents the combustion efficiency of the diesel engine, and $N_{D}$ is the rotational speed of the diesel engine.

(2) Fuel Injection Pump

The characteristics of the injection pump are represented by the injection volume $g_{cyl}$, which is a function of the displacement $F_{r}$ of the throttle rack and the speed $N_{D}$ of the diesel engine:(3)$$g_{cyl}=f\left(N_{D},F_{r}\right)$$

For a multicylinder diesel engine, its calculation can be calculated separately for each cylinder injection, and then its average value can be taken. However, the calculation is more complicated, so this paper gives an approximate formula according to the literature [22]:(4)$$\tau =15/N_{D}$$

(3) Intake air flow

The intake air flow $\dot{m}$ consists of 2 parts, the cylinder closed air flow ${\dot{m}}_{tr}$ and the swept air flow ${\dot{m}}_{SC}$.
(5)$$\dot{m}={\dot{m}}_{tr}+{\dot{m}}_{SC}$$
(6)$${\dot{m}}_{tr}=V\cdot N_{cy}\cdot \frac{N_{D}}{120}\cdot \eta_{\nu }\cdot \frac{P_{I}}{RT_{I}}$$
(7)$${\dot{m}}_{SC}=\frac{A_{\nu }\cdot P_{I}}{\sqrt{T_{I}}}{\left\{\frac{2K_{1}}{K_{1}-1}\frac{1}{R}\left[{\left(\frac{P_{E}}{P_{I}}\right)}^{\frac{2}{K_{1}}}-{\left(\frac{P_{E}}{P_{I}}\right)}^{\frac{K_{1}+1}{K_{1}}}\right]\right\}}^{\frac{1}{2}}$$
where $\eta_{\nu }$ is the inflation coefficient, $A_{v}$ is the swept volume coefficient, $N_{cy}$ is the number of diesel cylinders, $P_{I}$, $T_{I}$ denotes the intake pressure and temperature, and $P_{E}$ denotes the pre-turbine pressure.

(4) Intercooler

Diesel intake air temperature:(8)$$T_{I}=T_{C}\left(1-\varepsilon \right)+\varepsilon T_{W}$$
where ε is the efficiency of the intercooler, and $T_{W}$ is the cooling water temperature.

Diesel intake pressure:(9)$$P_{i}=P_{c}-\Delta P_{cool}$$
where $\Delta P_{cool}$ is the intercooler pressure loss.

(5) Pre-turbine temperature

The pre-turbine temperature $T_{E}$ (diesel exhaust temperature) determines the amount of exhaust energy, which is a function of the overall air-fuel ratio $A_{F2}$ and the rotational speed:(10)$$T_{E}=\Delta T_{E}+T_{i}$$
(11)$$\Delta T_{E}=\frac{x}{1+A_{F2}}\cdot \frac{\omega \eta_{cy}}{C_{pex}}+f\left(N_{D}\right)$$
where ω is the low calorific value of the fuel and *x* is the proportion of the energy carried into the exhaust pipe by the exhaust gas to the amount of fuel injected into the cylinder.

(6) Turbocharger
(12)$$\frac{\pi I_{tc}}{30}\cdot \frac{dN_{tc}}{dt}=M_{T}\eta_{MT}-\frac{M_{C}}{\eta_{MC}}$$
where $\eta_{MT}$ and $\eta_{MC}$ denote the mechanical efficiencies of the turbine and compressor, respectively.

#### 2.1.2. Diesel Engine Simulation Model Construction

In this study, the diesel engines of 6L21/31 Marine are modeled by utilizing the specialized diesel engine modeling software AVL-Boost R2020. According to the structural composition and parameters of the diesel engine shown in Table 1. The corresponding modules within the AVL-Boost software are used to represent each subsystem [23,24,25], and the simulation model of the whole engine is obtained as shown in Figure 2.

#### 2.1.3. Diesel Engine 3D Model Construction

In this section, we use SolidWorks 2017 modeling software to complete the visual modeling of the diesel engine with reference to the relevant data of the diesel engines of 6L21/31 Marine, and on this basis, we use the Animator plug-in to complete the simulation of the working process of the diesel engine [26]. The specific modeling steps are as follows:

(1) Obtain the information of each part of the diesel engine and establish the three-dimensional model of each part of the diesel engine;

(2) Correct the dimensions of each part model to meet the fit requirements;

(3) Determine the assembly order and assembly relationship of each component, and complete the assembly design;

(4) Complete the assembly work, and carry out the diesel engine operation simulation. The diesel engine visualization results are shown in the following Figure 3:

#### 2.1.4. Validation of Model Validity

To substantiate the precision of the developed model further, an assessment of the DT model’s effectiveness is conducted. This involves simulating the diesel engine under three distinct operational conditions: full load (100%), three-quarter load (75%), and half load (50%). The DT model’s performance is stabilized at each load level, and the corresponding outcomes are documented. Subsequently, the simulated results from the DT model are compared against the actual data. The comparative analysis is presented in Table 2, illustrating the model’s accuracy and reliability.

According to Table 2, the model’s simulated data under the three operational states closely match the actual data, with a maximum divergence of 3%. This result confirms that the developed DT model adheres to the stringent criteria for high fidelity.

#### 2.1.5. Fault Deduction

To address the problem of scarcity of fuel system fault data, this paper analyzes the causes of fuel system faults according to the literature [27], and then modifies the corresponding parameters in the DT model to carry out fault deduction. In this paper, based on the digital twin model, the fuel system normal state, supercharger failure, injection advance, injection lag, cooler failure, injector wear, and fuel supply pipe blockage are deduced to obtain the relevant fault data in seven health states. The specific derivation is shown in Table 3:

### 2.2. Key Theories

Owing to the limited information available on fault characteristics within small samples, coupled with the propensity of traditional neural networks to underfitting, a Siamese network architecture is used to expand the number of training. Subsequently, a ViT encoder is utilized as a subnetwork to extract global multiscale features from the input data, facilitating deep learning processes. The fault features thus extracted are then integrated into an improved KNN algorithm for fault classification. This section delves into the theoretical underpinnings of Siamese networks, the ViT encoder, and the KNN classifier.

#### 2.2.1. Siamese Networks

The idea of siamese network is to compare 2 groups of input samples, its structure is shown in Figure 4, the model firstly uses 2 subnetworks with shared weights to extract the features of the input sample $\left(X_{1},X_{2}\right)$, then calculates the distance between the 2 groups of features in the feature space, and measures the similarity of the 2 groups of samples by the distance between the sample features and outputs them [28]. When the 2 samples belong to the same category, the more the similarity label $Y_{i,j}$ of the model output tends to 1, and $Y_{i,j}$ tends to 0 if the 2 samples belong to different categories.

#### 2.2.2. Vision Transformer Encoder

The encoder is the main component of the Vision Transformer, which consists of a stack of *N* encoders with similar structures. Each encoder is in turn mainly composed of MultiHead Self-Attention (MSA) and MultiLayer perceptron (MLP) layers [29,30]. Among them, MSA, as the core component of the encoder, can compute the attention weights of the input vectors through multiple dimensions to extract multifaceted features, which has excellent global feature extraction capability. MSA consists of multiple single-head self-attention and the single-head self-attention mechanism is computed as shown below:(13)$$\left\{\begin{array}{c} Q=W_{q}\cdot T\\ K=W_{k}\cdot T\\ V=W_{\nu }\cdot T \end{array}\right.$$
(14)$$Attention\left(Q,K,V\right)=soft\max\left(\frac{QK^{T}}{\sqrt{d_{k}}}\right)V$$
where *Q*, *K*, *V* are three different weight matrices obtained by three different linear transformations of the input vector *T*, $\sqrt{d_{k}}$ denotes the dimensionality of *Q*, *K*, *V*, and Attention(·) is the self-attention computation.

MSA as an improved version of the single-head attention mechanism consists of self-attention, which allows the model to focus on several different locations from different subspaces to capture different levels of features and information, and finally the outputs of the *h* single-head attention mechanism are connected and then fused by the linear mapping weight matrix $W^{o}$ to obtain the final feature information, which is shown as follows:(15)$$head_{i}=Attention\left(Q_{i},K_{i},V_{i}\right)\;i=1,2\cdots h$$
(16)$$Multihead\left(Q,K,V\right)=Concat\left(head_{1},\ldots ,head_{h}\right)W^{o}$$
where $head_{i}$ is the output of *h* single-headed self-attentive mechanisms and Concat(·) is the splicing operation.

#### 2.2.3. K-Nearest Neighbor Classification Algorithm

The KNN algorithm is one of the most widely used classification algorithms proposed by Cover and Hart [31], with the advantages of fast computation speed and no need for training. Its principle is to find the *K* training samples in the feature space closest to the sample to be tested. If the vast majority of the *K* samples belong to a certain category, it is assumed that the sample to be tested also belongs to this category. The steps of KNN classification are as follows:

(1) Select the parameter *K*.

(2) Calculate the distance between the sample to be tested and all known samples.

(3) Select the *K* samples closest to the sample to be tested.

(4) Output the sample category to be tested as the most numerous category among the *K* nearest samples according to the voting law of minority to the majority.

### 2.3. Fault Diagnosis Method Based on SViT-KNN

#### 2.3.1. Data Preprocessing

When a fuel system operates, it inevitably generates various noises, such as mechanical noise and combustion noise. The noise will make the already limited data contain more misleading information with small samples, thus affecting the generalization ability of the model [32]. Especially in ViT encoder, the attention mechanism may be more sensitive to the noise in the samples, and the noise may be extracted as an important feature in the self-attentive computation, affecting the classification of the subsequent model. So, it is meaningful to denoise the samples. In this paper, the traditional wavelet threshold denoising method is used, which has sound localization and multiresolution characteristics and can effectively distinguish between data and noise; the steps are as follows:

(1) Wavelet decomposition: the noise-containing signal x(t) is decomposed to different frequencies by wavelet transform with the following equation:(17)$$W_{k}=DWT\left(x\left(t\right),\phi_{k}\right)$$
(18)$$W_{k}=A_{j}+\sum_{k=1}^{j}D_{k}$$
where $W_{k}$ denotes the decomposed wavelet signal, DWT denotes the discrete wavelet transform, $\phi_{k}$ is the wavelet basis function, $A_{j}$ is the low-frequency part of layer *j*, and $D_{k}$ is the high-frequency part of layer *k*. In this paper, $\phi_{k}$ chooses haar wavelet, and the number of decomposition layers *k* is taken as 3.

(2) Threshold processing: After the signal has undergone wavelet transformation, the distribution of the signal and noise on the spectrum is often different. The signal portion usually appears at low frequencies, while the noise primarily appears at high frequencies. An appropriate threshold is selected to filter out the abnormal frequency parts of the signal by utilizing this characteristic. The formula is as follows:(19)$${\hat{W}}_{k}=\left\{\begin{array}{cc} W_{k} & \mathrm{if}\;|W_{k}|>\lambda \\ 0 & \mathrm{otherwise} \end{array}\right.$$
where ${\hat{W}}_{k}$ is the denoised wavelet signal, λ is the threshold, the Sqtwolog threshold is selected as the wavelet threshold.

(3) Wavelet reconstruction: the thresholded wavelet signal is de-noised by inverse transformation reconstruction, and the reconstructed signal $\hat{x}\left(t\right)$ is obtained by the following equation:(20)$$\hat{x}\left(t\right)=IDWT\left({\hat{A}}_{j},{\left\{{\hat{D}}_{k}\right\}}_{k=1}^{j}\right)$$
where IDWT denotes the discrete wavelet inverse transform and $\hat{x}\left(t\right)$ is the denoised data.

To further improve the data quality under small sample sizes, the denoised data are finally subjected to maximum and minimum normalization to eliminate the influence of data dimensions. The resulting data are used as input for subsequent models.

#### 2.3.2. SViT Feature Extraction Network

In contemporary Siamese network designs, CNNs are predominantly used as subnetworks for feature extraction, with these networks adept at capturing local features of the input data via convolutional kernels of varying dimensions. Nevertheless, the constraints imposed by the receptive field size and the depth of the CNN model impede its ability to attend to the global features within the input data effectively. In contrast, the ViT encoder, leveraging its intrinsic attention mechanism, demonstrates a superior capacity to concentrate on the global characteristics of the input samples [33,34,35]. To concurrently extract both local details and global information from the data, a more sophisticated architecture termed the Siamese Vision Transformer is introduced. This architecture is integrated with an improved KNN classifier to facilitate the automatic classification of input samples. The structure of the proposed model is delineated in Figure 5, with the detailed parameters enumerated in Table 4.

After preprocessing, samples $X={\left[\begin{array}{c} x_{1},x_{2},\ldots ,x_{11} \end{array}\right]}^{T}$ and $Y={\left[\begin{array}{c} y_{1},y_{2},\ldots ,y_{11} \end{array}\right]}^{T}$ are transmitted to the SViT feature extraction network for the extraction of pertinent features. Within this network, the encoder embedding module translates the input samples into vectors of dimension 11×1, each encapsulating a set of thermal parameters relevant to the diesel fuel system. Subsequently, the model deploys a multihead self-attention mechanism to compute the self-attention of the input vectors concurrently. This mechanism enables each self-attention unit to concentrate on distinct aspects of the input vectors, facilitating the capture of a hierarchy of features and information. The output of the multihead self-attention is then directed to the multilayer perceptron layer, where it undergoes nonlinear transformations and mappings to yield the final encoder output. This output comprises a rich representation of the input data across multiple dimensions. Finally, the extracted features from the encoder are transmitted to the KNN classifier for automatic classification.

The Transformer encoder relies on its multiattention mechanism to extract features from the input data, which focuses on both global and local features of the data and makes the model better understand the correlation and importance of different features by calculating the attention weights of the features in each dimension, which effectively avoids the shortcomings of the local feature extraction and provides a convenient method of subsequent data classification.

#### 2.3.3. Weighted-Similarity Loss Function

Due to the complexity and diversity of fuel system faults, the values of thermal parameters of different types of faults may be closer. Traditional loss functions, during the training of models, primarily concentrate on the relationship between model outputs and labels, neglecting the connections between sample categories. Leading to suboptimal performance when classifying samples of similar categories. This study introduces a novel Weighted-Similarity loss function to bolster further the model’s capability to discern between identical category samples. By incorporating similarity labels and penalty coefficients, the model can effectively distinguish between samples belonging to similar categories. This enhanced loss function significantly elevates the precision in distinguishing between similar and dissimilar sample pairs, with its composition comprising two distinct components: similarity loss and penalty loss.

(1) Loss of similarity: for pairs of similar samples (the similarity label $Y_{i,j}$ is 1), the loss function calculates the square of the Euclidean distance, which encourages the model to reduce the distance between the features of the similar samples by the following formula:(21)$$L_{S}=\sum_{i,j=1,i\neq j}^{N}\left[Y_{i,j}{\left(1-R_{i,j}\right)}^{2}\right]$$

(2) Penalty Loss: For dissimilar sample pairs (the similarity label $Y_{i,j}$ is 0), he loss function calculates a penalty loss. The penalty coefficient is determined based on the difficulty of distinguishing between different categories of sample pairs. The penalty loss incentivizes the model to augment the feature distance between samples of distinct categories beyond a predefined threshold, as per the following formula:(22)$$L_{S}=\sum_{i,j=1,i\neq j}^{N}\left[\alpha_{i,j}\left(1-Y_{i,j}\right){\left(0-R_{i,j}\right)}^{2}\right]$$

Integrating the aforementioned two components, the final formula for the Weighted-Similarity loss function is as follows:(23)$$L_{S}=\sum_{i,j=1,i\neq j}^{N}\left[Y_{i,j}{\left(1-R_{i,j}\right)}^{2}+\alpha_{i,j}\left(1-Y_{i,j}\right){\left(0-R_{i,j}\right)}^{2}\right]$$
where $Y_{i,j}$ denotes the similarity label between the two samples and $\alpha_{i,j}$ denotes the penalty coefficient when sample *i* and sample *j* belong to different fault types. When the two samples belong to different categories, if the $R_{i,j}$ value of the network output is not close to zero, a larger $\alpha_{i,j}$ value is assigned to increase the loss During the training process the loss value can motivate the model to learn more accurately the subtle differences between pairs of samples with similar categories. After determining the loss function the model is trained using the gradient descent method, the goal of training is to make the samples of the same category as close to the feature distance as possible, while the samples of different categories are as far away from the feature distance as possible, so as to significantly improve the accuracy of the model to classify samples of similar categories.

#### 2.3.4. Improved KNN Classifier

Traditionally, the KNN algorithm classifies samples solely based on their distances to all training samples, which are computed across all features. Among these features, some are strongly correlated with the classification task, others are weakly correlated, and others are entirely unrelated [36]. In light of this, this paper introduces an attention mechanism within the KNN search process. This mechanism allows the model to extend beyond mere distance-based classification, leveraging the attention mechanism to weigh the neighbor voting in the KNN framework impartially. By doing so, the model moves away from a simplistic majority voting approach, thereby enhancing the accuracy of the classification process. The operational steps are outlined as follows:

(1) Find the *k* nearest neighbors $\left\{\begin{array}{c} x_{1},x_{2},\ldots x_{k} \end{array}\right\}$ of the sample *x* to be tested based on the Euclidean distance with the following equation:(24)$$d_{f}^{2}\left(x_{1},x_{2}\right)=\sqrt{\sum_{j=1}^{n}{\left(f_{j}\left(x\right)-f_{j}\left(x_{i}\right)\right)}^{2}}$$
where $f_{j}$ is the number of features for *j* samples and *n* is the total number of features.

(2) Calculate the weight $\alpha_{i}$ of each neighbor $x_{i}$ on *x* through the attention mechanism, where the Key, Value, Query relationship is as follows. First, use the dot product of data sample K and query Q to calculate the similarity between the samples with the following formula:(25)$$Similarity\left(x_{i},x\right)=Q^{T}K_{i}$$

Then, the obtained similarity is normalized to obtain the attention coefficient with the following formula:(26)$$s_{i}=soft\max\left(Similarity\left(x_{i},x\right)\right)$$

Finally, the weight $\alpha_{i}$ of each nearest neighbor $x_{i}$ to *x* is obtained by the dot product operation of the attention coefficients and the attention value Value, which is calculated as follows:(27)$$\alpha_{i}=\sum_{i=1}^{k}s_{i}\cdot x_{i}$$

(3) These weights are then used to calculate the weighted voting result *V*, which is calculated as follows:(28)$$V=\sum_{i=1}^{k}\alpha_{i}\cdot l_{i}$$
where $l_{i}$ is the category of the *i* neighbor.

Finally, the category of *x* is determined based on the weighted vote *V*. The highest weighted vote is chosen as the category with the following formula:(29)$$\hat{y}=\arg\max_{l}V_{l}$$
where $\hat{y}$ is the final category label, which is the maximum value chosen based on the weighted vote *V*.

#### 2.3.5. Overall Diagnostic Process

Figure 6 illustrates the comprehensive diagnostic workflow for the ship’s fuel system under conditions of few available samples. The detailed steps are as follows:

(1) Data Acquisition: fault simulation is conducted to generate fault data utilizing the established diesel engine DT model.

(2) Data Preprocessing: raw data are standardized by normalizing the maximum and minimum values to eliminate the influence of the scale.

(3) Model Training: Preprocessed data are segmented into training and testing subsets. From the training set. Two samples are randomly selected from the training set to form pairs, which are then input into the Siamese network. The loss function is computed by Equation (Equation 19), and the model’s parameters are updated using the gradient descent method. Training concludes when the model’s accuracy meets the predefined threshold or the maximum number of training iterations is reached, with the model being saved at this juncture.

(4) Fault Diagnosis: Input sample pairs are randomly generated from the test set and fed into the trained model. Following feature extraction, the classifier autonomously generates diagnostic outcomes.

(5) Output the diagnostic results and troubleshoot the diesel engine based on the results.

## 3. Experimental Validation

In this section, ablation and comparison experiments are set up, and the results are analyzed to demonstrate the effectiveness of the proposed method. All the experiments are run in the same environment, the deep learning framework is built by Pytorch 2.0.1, and the computers are configured as follows: the GPU is NVIDIA GeForce RTX 3060, and the CPU is Intel Core i7 12700H. Three sets of experiments are conducted, with the number of samples per class participating in the network training, *n*, set at 10, 20, and 30 for each set of experiments, respectively. The model’s accuracy, precision, recall, and F1 score are selected as the evaluation indexes. Each group of experiments is carried out 10 times, respectively, to ensure the reliability of the experimental results, and the final results are averaged for comparison.

### 3.1. Ablation Experiments

This section validates the effectiveness of the method proposed in this study by adding specific steps to the Siamese Convolutional Neural Network (SCNN) model from the literature [37] and comparing the experimental results.

(1) SCNN, which uses Siamese Convolutional Neural Networks for feature extraction and FC layer-based classification, and loss function using traditional cross-entropy loss function.

(2) SViT-KNN, using ViT encoder instead of CNN for feature extraction, improved KNN for classification and loss function using traditional cross-entropy loss function.

(3) The diagnostic model of the method proposed in this paper uses SViT-KNN, and the loss function uses the Weighted-Similarity loss function. The SCNN model structure and parameters are shown in Figure 7:

The experimental outcomes are presented in Table 5, alongside Figure 8 and Figure 9, which collectively assess the performance of the three methods across varying sample sizes using metrics such as accuracy, precision, F1 score, and recall.

Observations indicate that the SCNN model achieves an accuracy of 94.6% when the sample count is substantial. However, as the number of samples engaged in training diminishes, the model’s accuracy experiences a steep decline. Particularly noteworthy is the accuracy of 83.63% when the training sample count per class is reduced to 10, suggesting that with a lack of samples, the CNN-based model struggles to extract a sufficient quantity of informative features, subsequently compromising its performance.

The SViT-KNN approach adopts the SViT network to extract global multiscale data features, which are then inputted into the KNN classifier for classification. In contrast to the SCNN model across all three experiments, SViT-KNN demonstrates a marked enhancement in accuracy, attributed to the superior feature extraction ability of SViT over SCNN. Nevertheless, both models exhibit many misclassifications when differentiating samples within similar categories, as evidenced in Figure 9. This is a consequence of the traditional loss function’s exclusive emphasis on the discrepancy between model outputs and labels, disregarding the categorical relationships between samples. Consequently, errors are prevalent when classifying samples with overlapping characteristics, thus hindering the potential for further augmentation in model performance.

The methodology introduced in this paper uses the SViT-KNN fault diagnosis model, augmented by the Weighted-Similarity loss function outlined in Section 2.3.3 for training. From Figure 8 and Figure 9, the proposed method in this paper not only performs stably but also has the highest accuracy, precision, F1 score, and recall in the three sets of experiments. Remarkably, even when the training sample count is reduced to 10, the method maintains an accuracy of 97.97%. Furthermore, the examination of Figure 9c,f,i illustrates a marked enhancement in the model’s capability to discern between samples of similar categories, underscoring the efficiency of the loss function devised in this paper in bolstering diagnostic accuracy.

In summary, the ablation experiments (1) and (2) substantiate that the integration of the Siamese network and ViT as a feature extractor outperforms SCNN when training samples are scarce. The cumulative evidence from (1), (2), and (3) corroborates that the loss function designed in this paper surpasses the traditional cross-entropy loss function in differentiating similar categories of samples, thereby enhancing the accuracy rate. Consequently, the methodologies and steps outlined herein are indispensable and efficacious.

### 3.2. Comparative Experiments

This section presents a comparative analysis to substantiate the proposed method’s superiority in the context of few-shot fault diagnosis, aligning the method with 1D-CNN classical methods, as well as a method outlined in [15,38].

(1) 1D-CNN is a classical deep learning network; the specific structure and parameters of the 1D-CNN model selected in this paper are shown in Figure 10.

(2) DCSN-DRN is another kind of Siamese network proposed in the [38] for marine engine fault diagnosis with small samples. The Sianese network was built by two Deep Residual Networks (DRNs) with shared weights, then it was used to learn discriminative information from limited fault samples. What’s more, the Siamese network applies strong forces to hard samples towards their corresponding correct distribution areas. Finally, the distribution area of intra-class samples was shrunk and further improved the accuracy of the classification.

(3) OSLNet is a generalized few-shot fault classification network proposed by [15], which designs an orthogonal Softmax layer on top of CNN as the classifier of the network, so that the model maximizes the difference between samples during training and testing, and the structure of the network is simplified.

The 1D-CNN and OSLNet model structure and parameters are shown in Figure 10:

Table 6 and Figure 11 and Figure 12 present the diagnostic results of each method in the comparative experimentation. When the quantity of training samples is adequate, the accuracy of all fault diagnosis approaches surpasses 90%. However, as the sample count in the training set diminishes, the diagnostic accuracy of the 1D-CNN and OSLNet networks experiences a drastic decline. Particularly, the 1D-CNN yields the poorest results, which can be attributed to its classification within the domain of traditional deep learning networks. This classification renders the 1D-CNN highly susceptible to the number of training samples, posing challenges in extracting sufficient features for learning under limited sample conditions.

Despite improved overall accuracy, DCSN-DRN and OSLNet show higher misclassification within similar categories, indicating a struggle to differentiate under limited samples. In contrast, the method presented in this paper demonstrates a remarkable resilience to variations in the number of training samples, achieving the highest performance across all four evaluation indexes in the three experimental groups. This comparative analysis underscores the substantial advantage of the proposed method in the diagnosis of diesel engine fuel system faults under conditions of small sample availability.

## 4. Conclusions

This paper proposes a DT-SViT-KNN few-shot fault diagnosis method to realize accurate and efficient fault diagnosis. Firstly, a DT model of the diesel engine is established, and the fault data are obtained through model deduction. Then, a new Siamese network was designed by combining the Siamese network with ViT to extract the global multiscale features of the input data. The Weighted-Similarity loss function is also designed to improve the model’s differentiation accuracy for similar samples to a new level. Finally, an improved KNN classifier based on the attention mechanism is added to the traditional Siamese network, enabling the model to classify and improve the diagnostic efficiency automatically. Ablation experiments and comparison experiments were conducted successively to verify that DT-SViT-KNN is an accurate and reliable few-shot fault diagnosis method.

However, there are still some challenges in practical application scenarios. Fault signals may be interfered with by a variety of noises, and the levels of these noises are usually unknown, which makes it difficult to select the number of wavelet decomposition layers and thresholds in wavelet denoising and affects the denoising effect. On the other hand, as the operating conditions and time of the system change, the data will become diversified, and the difference in the distribution of the same type of data will become larger, so how to carry out fault diagnosis under variable operating conditions with small samples is also a difficult problem. In future research work, we will investigate the performance of the proposed method on real data and further solve the problems encountered in the practical application of this method.

## Figures and Tables

**Figure 1 sensors-25-00006-f001:**
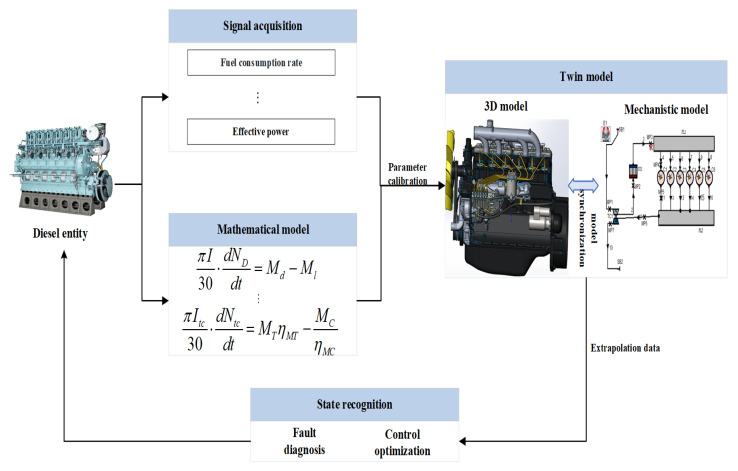
Composition of diesel engine digital twin system.

**Figure 2 sensors-25-00006-f002:**
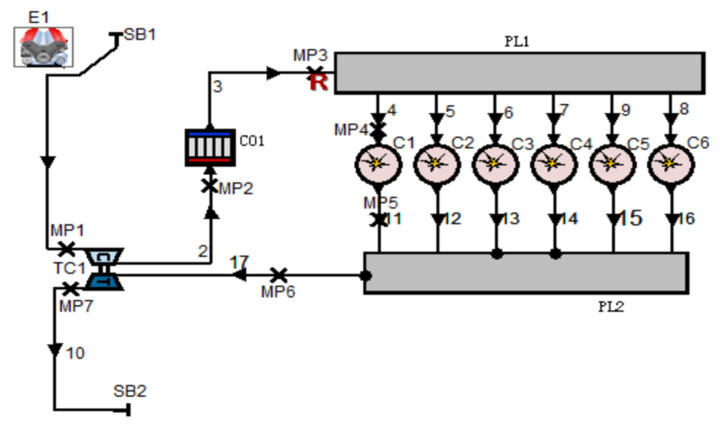
Simulation schematic of the 6L21/31 marine diesel engine.

**Figure 3 sensors-25-00006-f003:**
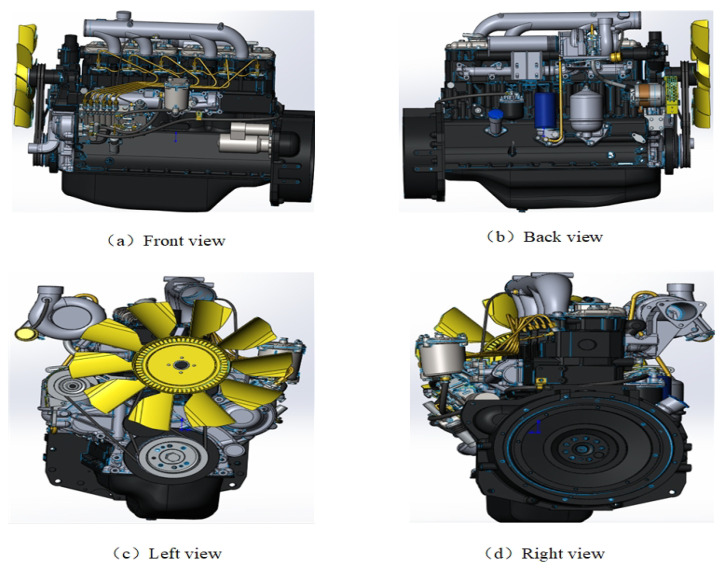
Twin model of 6L21/31 marine diesel engine.

**Figure 4 sensors-25-00006-f004:**
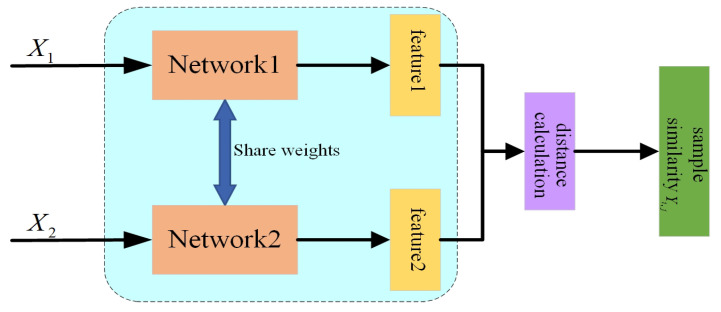
Structure of the siamese network.

**Figure 5 sensors-25-00006-f005:**
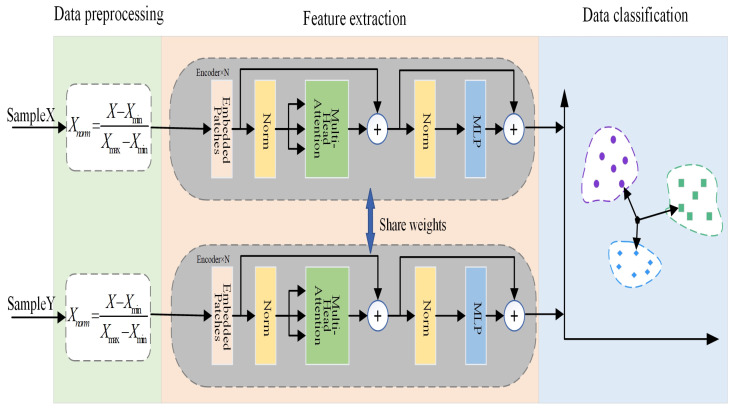
Structure of SViT-KNN model.

**Figure 6 sensors-25-00006-f006:**
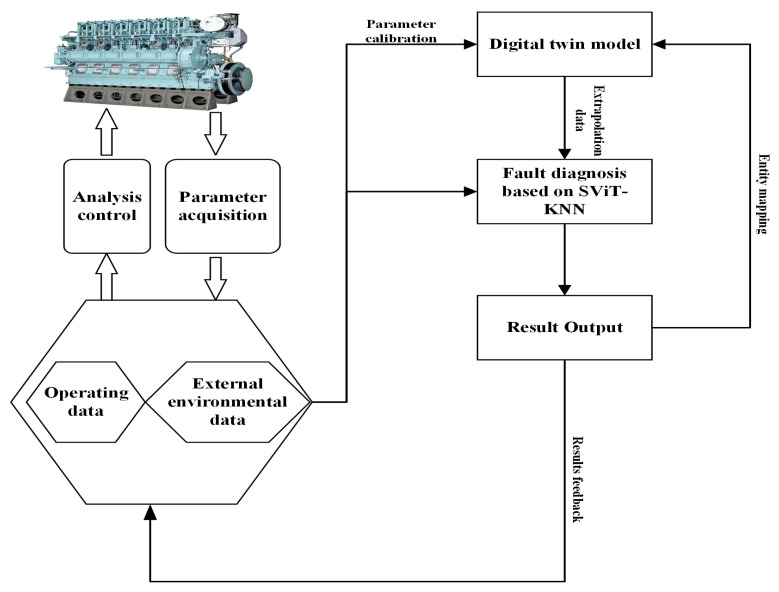
Overall Diagnostic Flowchart.

**Figure 7 sensors-25-00006-f007:**
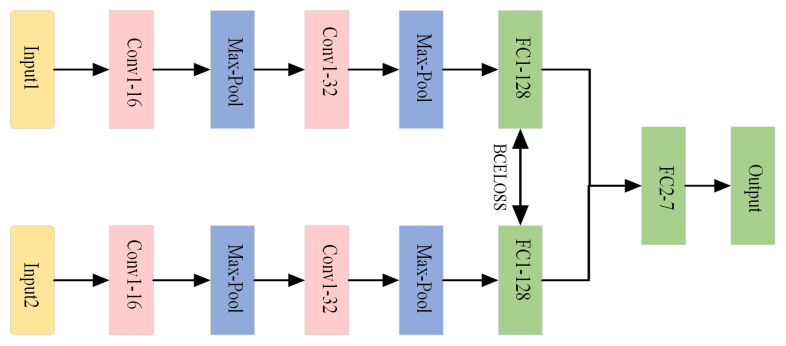
Structure of SCNN model.

**Figure 8 sensors-25-00006-f008:**
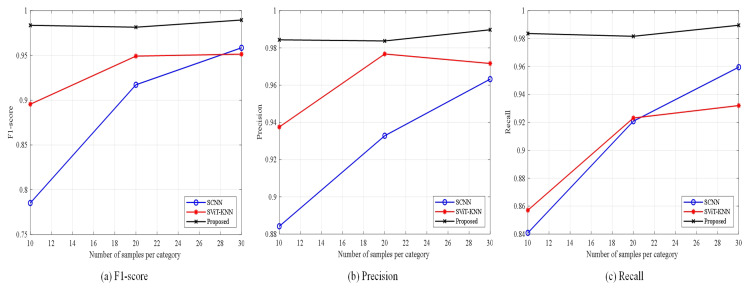
Experimental results of the three methods with different sample sizes.

**Figure 9 sensors-25-00006-f009:**
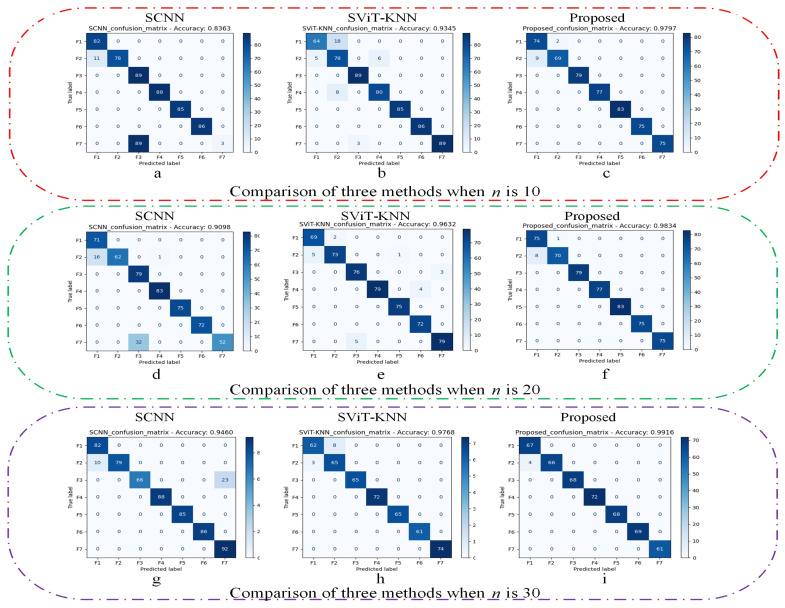
Confusion matrix of the three methods for different sample sizes.

**Figure 10 sensors-25-00006-f010:**
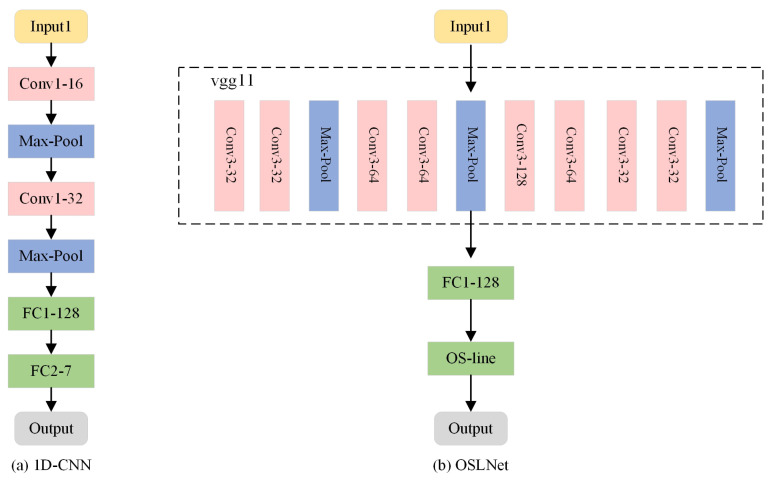
Structure of 1D-CNN and OSLNet models.

**Figure 11 sensors-25-00006-f011:**
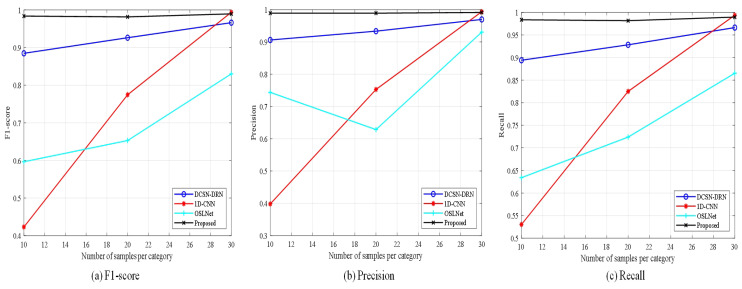
Experimental results of the four methods with different sample sizes.

**Figure 12 sensors-25-00006-f012:**
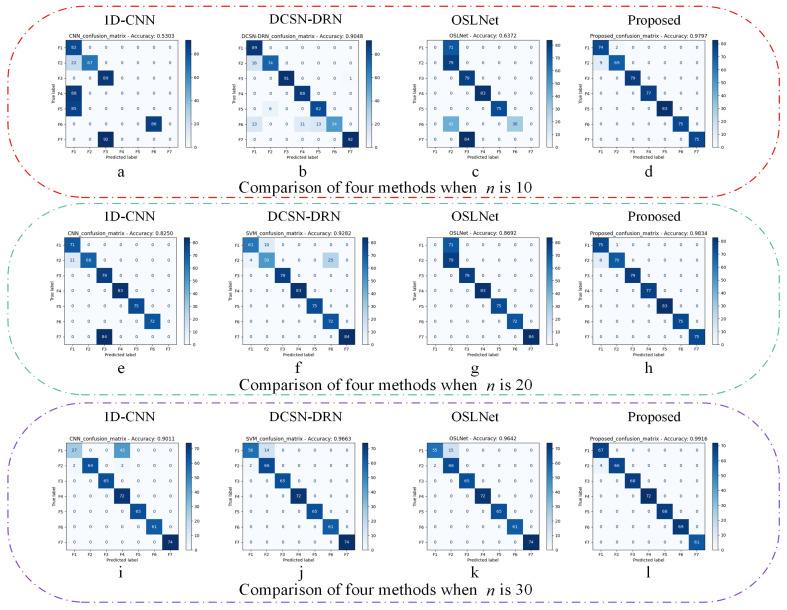
Confusion matrix of the four methods for different sample sizes.

**Table 1 sensors-25-00006-t001:** Main parameters of 6L21/31 marine diesel engine.

Items	Parameters
Number of cylinders	6
Stroke count	4
Bore (mm)	210
Piston stroke (mm)	310
Compression ratio	15.5
Rotation speed (r/min)	900
Cylinder displacement (dm^3^)	10.73
Mean effective pressure (bar)	24.6
Fuel consumption rate (g/(kW · h))	195
Maximum combustion pressure (bar)	200
Rated power (kW)	1290

**Table 2 sensors-25-00006-t002:** Comparative Analysis of Deduced Data and Actual Data.

Parameters	Operating Conditions								
	100%			75%			50%		
	Real Value	Simulation Value	Errors	Real Value	Simulation Value	Errors	Real Value	Simulation Value	Errors
Fuel consumption rate g/kW · h	190.5	187.73	1.45%	186.5	184.97	0.82%	191.9	188.52	1.76%
Effective power kW	1200	1190.91	0.76%	900	888.25	1.31%	600	597.19	0.47%
Combustion pressure bar	197.4	202.35	2.51%	161.4	160.52	0.55%	120.8	117.37	2.84%
Mean effective pressure bar	23.93	24.65	3%	18.02	18.45	2.39%	12.03	12.32	2.41%

**Table 3 sensors-25-00006-t003:** Types of faults and the method of deduction.

Fault Type	Method of Deduction	Fault Label
Normal state	Normal system operation	F1
Turbocharger failure	Adjust turbocharger pressure efficiency	F2
Fuel injection advance	Adjust the injection angle	F3
Fuel injection lag	Adjust the injection angle	F4
Cooler failure	Adjust cooler efficiency	F5
Injector wear	Adjust the cylinder oil supply volume	F6
Clogged oil supply lines	Adjust the cylinder oil supply volume	F7

**Table 4 sensors-25-00006-t004:** Main parameters of the model.

Network	Model Parameters	Value
	Input data dimensions	11 × 1
Feature extraction network	Batch size	64
	Learning rate	0.001
	Transformer code block	2
	Number of multihead attention	8
	Embedding dimensions	512
	Hidden dimensions	512
Classifier	Nearest neighbor K	1

**Table 5 sensors-25-00006-t005:** Comparison of the three methods at different sample sizes.

Sample Size *n*	Indexs	Methods		
		SCNN	SViT-KNN	Proposed
10	Accuracy	0.8363	0.9345	**0.9722**
	Recall	0.8409	0.8571	**0.9836**
	Precision	0.8842	0.9375	**0.9843**
	F1	0.7852	0.8955	**0.9836**
20	Accuracy	0.9098	0.9632	**0.9821**
	Recall	0.9208	0.9231	**0.9816**
	Precision	0.9328	0.9767	**0.9837**
	F1	0.9172	0.9492	**0.9815**
30	Accuracy	0.9460	0.9768	**0.9913**
	Recall	0.9595	0.9320	**0.9895**
	Precision	0.9632	0.9716	**0.9897**
	F1	0.9585	0.9514	**0.9895**

**Table 6 sensors-25-00006-t006:** Comparison of the four methods at different sample sizes.

Sample Size *n*	Indexs	Methods			
		1D-CNN	DCSN-DRN	OSLNet	Proposed
10	Accuracy	0.5303	0.9023	0.6372	**0.9722**
	Recall	0.5303	0.8940	0.6339	**0.9836**
	Precision	0.3978	0.9062	0.7438	**0.9894**
	F1	0.4232	0.8847	0.5966	**0.9836**
20	Accuracy	0.8250	0.9246	0.8692	**0.9821**
	Recall	0.8250	0.9282	0.7238	**0.9816**
	Precision	0.7528	0.9335	0.6285	**0.9893**
	F1	0.7745	0.9261	0.9815	**0.9815**
30	Accuracy	0.9011	0.9676	0.9642	**0.9913**
	Recall	0.9937	0.9663	0.8653	**0.9895**
	Precision	0.9939	0.9699	0.9306	**0.9917**
	F1	0.9937	0.9661	0.9895	**0.9895**

## Data Availability

Data are contained within this article.

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
