# Peer review of "Study on Few-Shot Fault Diagnosis Method for Marine Fuel Systems Based on DT-SViT-KNN"

_sensors, 2024, doi:10.3390/s25010006_

Round 1

Reviewer 1 Report

Comments and Suggestions for Authors

This paper proposes a fault diagnosis method for marine fuel systems using Digital Twin, Siamese Vision Transformer, and K-Nearest Neighbor. It constructs a DT model for fault simulation, combines SViT for feature extraction, and enhances classification with an attention-based KNN and a Weighted-Similarity loss function. This is an interesting engineering paper, but before considering acceptance, it requires thorough revision.

1.     The author should clarify the definition of Digital Twin. The model used in this paper resembles a simulation model rather than a Digital Twin model.

2.     The formatting throughout the paper requires attention. For instance, the "where" used to introduce the parameters in the equations should be aligned to the left and kept in lowercase.

3.     How does the model proposed by the author address noise?

4.     The references should be revised to include recent literature from the past two years, particularly from this journal and related journals. Additionally, citations of non-essential sources, e.g., Reference 3, should be avoided.

5.     The comparative experiments are incomplete, as the models compared are too few and outdated. At least three recent models from the past two years should be added for comparison.

6.     Finally, the outlook on future work is vague and lacks substance.

Comments on the Quality of English Language

The language in this paper needs improvement, as it currently reads like a direct translation from Chinese using a translation tool.

Author Response

We feel great thanks for your professional review work on our article. As you are concerned, there are several problems that need to be addressed.According to your nice suggestions, we have made extensive corrections to our previous draft, the detailed corrections are listed below.

Comments 1:[The author should clarify the definition of Digital Twin. The model used in this paper resembles a simulation model rather than a Digital Twin model.]

Response 1: [Thank you for pointing this out. We think this is an excellent suggestion. We have redefined the second paragraph on page 1 to elaborate the application of digital twins in fault diagnosis. Similar to references [4],[5],[6],[7],[8], DT is mainly used in this paper for fault extrapolation to obtain fault data, and the specific fault deduction method is given in the newly added section 2.1.5.]

Comments 2:[The formatting throughout the paper requires attention. For instance, the "where" used to introduce the parameters in the equations should be aligned to the left and kept in lowercase.]

Response 2: [We feel sorry for our carelessness. In our resubmitted manuscript, the related errors on page 4, page 5, page 8, page 9, page 10 is revised. Thanks for your correction.]

Comments 3:[How does the model proposed by the author address noise?]

Response 3: [Thank you for pointing this out. We agree with this comment. Therefore, we have added a new section 2.3.1 on page 9 to explained in detail the method that used to denoise.]

Comments 4:[The references should be revised to include recent literature from the past two years, particularly from this journal and related journals. Additionally, citations of non-essential sources, e.g., Reference 3, should be avoided.]

Response 4: [We totally agree with the reviewer, as suggested by the reviewer. We have added relevant references, particularly from this journal and related journals. As seen in page 1/line 7 /Section 1, page 1/line 13 /Section 1, page 2/line 1 /Section 1, page 2/line 12 /Section 1, page 2/line 12 /Section 1, page 2/line 18 /Section 1, page 6/line 4 /Section 2.1.3, page 7/line 2 /Section 2.1.5, page 8/line 3 /Section 2.2.2, page 9/line 18 /Section 2.3.1. We also have deleted non-essential sources. ]

Comments 5:[The comparative experiments are incomplete, as the models compared are too few and outdated. At least three recent models from the past two years should be added for comparison.]

Response 5: [Thank you for your valuable feedback on our manuscript. We fully understand the need to strengthen our discussion and experimental sections. As suggested by the reviewer, We revised the comparison experiment section, and chose the method of reference [15],[39] to compare with the method proposed in this paper, especially the literature [39] is consistent with the research object and research background of this paper, which verifies the superiority of the method proposed in this paper.]

Comments 6:[Finally, the outlook on future work is vague and lacks substance.]

Response 6: [Thanks for the correction. We have added a discussion in Section 4  about the various noise problems of implementing DT-SViT-KNN in real-world fault diagnosis scenarios and the problem that under variable operating conditions with small samples, Additionally, In a vision for future work, we will explore the performance of the proposed method on real datasets and how it addresses the above issues. ]

Reviewer 2 Report

Comments and Suggestions for Authors

This paper presents DT-SViT-KNN, a novel fault diagnosis method combining a Digital Twin (DT) model, a Siamese Vision Transformer (SViT), and an attention-enhanced KNN classifier. The approach effectively handles small-sample fault diagnosis by generating simulated fault data through the DT model and improving classification accuracy with a Weighted-Similarity loss function. The method achieves high accuracy (up to 99.16%) with limited training samples, offering strong innovation and practical applicability. However, there are still some areas for improvement:  

1. It would be helpful if the authors could provide more technical details about the fault data generation process described in Section 2.1.2 (Diesel Engine Simulation Model Construction). While the paper mentions that fault simulation is conducted using the DT model, the specific steps or mechanisms for generating fault data are not entirely clear. For instance, it would be valuable to know what types of faults were simulated, how the parameters were varied, and whether any real-world fault data were used for validation. Including this information would enhance the reproducibility of the study and help readers better understand the practical implementation of the proposed method.  

2. You could cite this work to enrich the introduction section of your paper: [1] Liu C, Chen Y, Xu X, Che W. Domain generalization-based damage detection of composite structures powered by structural digital twin. Composites Science and Technology. 2024;258:110908.

3. To enhance the practical relevance of the study, the authors could consider adding a brief discussion in Section 4 (Conclusion) about the potential challenges of implementing DT-SViT-KNN in real-world fault diagnosis scenarios. For example, addressing how the method might handle noisy or incomplete data, which are common in practical applications, or how it could be adapted to larger and more diverse datasets, would provide valuable insights for readers. Additionally, suggesting future work, such as testing the method in real-time monitoring systems or validating it on real-world datasets, could further highlight its practical potential and inspire follow-up research.

Comments on the Quality of English Language

NA

Author Response

Thank you for your careful review of our manuscript. We have taken the reviewers"comments into serious consideration and made significant revisions to the paper. We have addressed all the issues raised and revised the corresponding sections accordingly: We believe that the manuscript has been lmproved significantly and now meets the journal's standards.

Comments 1:[It would be helpful if the authors could provide more technical details about the fault data generation process described in Section 2.1.2 (Diesel Engine Simulation Model Construction). While the paper mentions that fault simulation is conducted using the DT model, the specific steps or mechanisms for generating fault data are not entirely clear. For instance, it would be valuable to know what types of faults were simulated, how the parameters were varied, and whether any real-world fault data were used for validation. Including this information would enhance the reproducibility of the study and help readers better understand the practical implementation of the proposed method.]

Response 1: [Thank you for pointing this out. We agree with this comment. Therefore, We have added a 2.1.5 Fault Extrapolation section that explain in detail what kinds of faults are deducted by the DT model and how they are deducted.]

Comments 2:[You could cite this work to enrich the introduction section of your paper: [1] Liu C, Chen Y, Xu X, Che W. Domain generalization-based damage detection of composite structures powered by structural digital twin. Composites Science and Technology. 2024;258:110908.]

Response 2: [We sincerely appreciate the valuable comments. We read this article carefully and found it to be very enlightening for our paper, and cited it on page 1/line 13 /Section 1.]

Comments 3:[To enhance the practical relevance of the study, the authors could consider adding a brief discussion in Section 4 (Conclusion) about the potential challenges of implementing DT-SViT-KNN in real-world fault diagnosis scenarios. For example, addressing how the method might handle noisy or incomplete data, which are common in practical applications, or how it could be adapted to larger and more diverse datasets, would provide valuable insights for readers. Additionally, suggesting future work, such as testing the method in real-time monitoring systems or validating it on real-world datasets, could further highlight its practical potential and inspire follow-up research.]

Response 3: [Thank you very much for your substantive comments! We have added a discussion in Section 4  about the various noise problems of implementing DT-SViT-KNN in real-world fault diagnosis scenarios and the problem that under variable operating conditions with small samples, Additionally, In a vision for future work, we will explore the performance of the proposed method on real datasets and how it addresses the above issues.]

Round 2

Reviewer 1 Report

Comments and Suggestions for Authors

The revised manuscript shows considerable improvement and can be considered for acceptance.